# Novel Scintillating Nanoparticles for Potential Application in Photodynamic Cancer Therapy

**DOI:** 10.3390/pharmaceutics14112258

**Published:** 2022-10-22

**Authors:** Bianca A. da Silva, Michael Nazarkovsky, Helmut Isaac Padilla-Chavarría, Edith Alejandra C. Mendivelso, Heber L. de Mello, Cauê de S. C. Nogueira, Rafael dos S. Carvalho, Marco Cremona, Volodymyr Zaitsev, Yutao Xing, Rodrigo da C. Bisaggio, Luiz A. Alves, Jiang Kai

**Affiliations:** 1Chemistry Department, Pontifical Catholic University of Rio de Janeiro, 225 Marquês de São Vicente Str., Rio de Janeiro 22451-900, Brazil; 2Laboratory of Cellular Communication, Oswaldo Cruz Institute, Fiocruz, 4365 Brasil Av., Manguinhos, Rio de Janeiro 21040-360, Brazil; 3Biotechnology Department, Federal Institute of Rio de Janeiro, 121 Senador Furtado Str., Maracanã, Rio de Janeiro 20270-021, Brazil; 4High-Resolution Electron Microscopy Lab, Advanced Characterization Center for Petroleum Industry (LaMAR/CAIPE), Fluminense Federal University, Niteroi 24210-346, Brazil; 5Physics Department, Pontifical Catholic University of Rio de Janeiro, 225 Marquês de São Vicente Str., Rio de Janeiro 22451-900, Brazil

**Keywords:** nanoparticles, rare earth oxides, photodynamic therapy, cancer, nanosilica

## Abstract

The development of X-ray-absorbing scintillating nanoparticles is of high interest for solving the short penetration depth problem of visible and infrared light in photodynamic therapy (PDT). Thus, these nanoparticles are considered a promising treatment for several types of cancer. Herein, gadolinium oxide nanoparticles doped with europium ions (Gd_2_O_3_:Eu^3+^) were obtained by using polyvinyl alcohol as a capping agent. Hybrid silica nanoparticles decorated with europium-doped gadolinium oxide (SiO_2_-Gd_2_O_3_:Eu^3+^) were also prepared through the impregnation method. The synthesized nanoparticles were structurally characterized and tested to analyze their biocompatibility. X-ray diffraction, scanning electron microscopy, and transmission electron microscopy confirmed the high crystallinity and purity of the Gd_2_O_3_:Eu^3+^ particles and the homogeneous distribution of nanostructured rare earth oxides throughout the fumed silica matrix for SiO_2_-Gd_2_O_3_:Eu^3+^. Both nanoparticles displayed stable negative ζ-potentials. The photoluminescence properties of the materials were obtained using a Xe lamp as an excitation source, and they exhibited characteristic Eu^3+^ bands, including at 610 nm, which is the most intense transition band of this ion. Cytotoxicity studies on mouse glioblastoma GL261 cells indicated that these materials appear to be nontoxic from 10 to 500 μg·mL^−1^ and show a small reduction in viability in non-tumor cell lines. All these findings demonstrate their possible use as alternative materials in PDT.

## 1. Introduction

With improved feeding and sanitary conditions and health treatments in most developing countries, human life expectancy has increased. On the other hand, cancer rates have also increased; thus, cancer has overcome cardiovascular disease as the main cause of death worldwide [1,2,3]. In this scenario, new treatments against cancer, such as photodynamic therapy (PDT), may mitigate these high death ratios.

PDT is defined as a noninvasive treatment strategy based on the administration of a photosensitizer, a substance that can be activated by a light source, resulting in the generation of reactive oxygen species (ROS) to eventually cause the death of cancer cells [4,5,6,7]. However, most photosensitizers (PSs) still display limitations concerning the treatment of deeper tumors because most PSs exhibit maximum absorption peaks in the UV–Vis region; these peaks overlap those in the absorption spectrum of human tissues [5,7,8]. An ingenious alternative is the use of scintillating nanoparticles that are capable of converting X-ray radiation to UV–Vis light, thus activating the applied PSs [5,7,9,10]. Examples of scintillating nanoparticles based on rare earth elements (REEs) functionalized with PSs have been reported in the literature in different biological studies [11,12]. The nanoparticles synthesized and characterized in this work are capable of emitting light when irradiated with X-rays. X-rays penetrate tissues well, and when a low-intensity source is used, the treatment risks are minimized. Using our nanoparticles conjugated to one or more photosensitizers already described in the literature could promote the treatment of deeper tumors in photodynamic therapy because a conventional PDT irradiator would not reach the tumor without probe or surgery.

In this scenario, REEs applied to the synthesis of nanoparticles offer the possibility to create new therapeutic tools that must be tested concerning their biocompatibility and photodynamic cancer cell activity, both in vitro and in vivo [13,14,15,16,17,18,19]. The present study aimed to develop a synthesis of gadolinium (III)-based nanoparticles doped with europium (III) through the sol–gel method. Additionally, we undertook the deposition of both elements onto a nanosilica surface (fumed silica with a specific surface area *S_BET_* > 300 m^2^/g) for controlling the size of the nanoparticles. Furthermore, optical and mechanical neutrality alongside the availability of the entire surface on fumed SiO_2_ allowed the crystalline phases of Gd^3+^ or Eu^3+^ to be uniformly distributed over the inert template, maintaining the pure optical property characteristics of both REE crystalline phases [20]. The impregnation of the nanosilica with REE salts and the application of such a hybrid system in PDT is an innovative step. The literature reveals identical compositions (pure/doped Gd_2_O_3_-SiO_2_); varying morphologies and/or fabrication procedures, e.g., mesoporous silica [21,22,23,24,25]; core–shell spheres obtained by laser ablation [26,27]; films or layers [28,29,30] in the form of single Gd_2_SiO_5_ phase nanoparticles [31]; and, eventually, silicas modified by Gd/Eu complexes [32].

When synthesizing new hybrid materials, toxicity is also a crucial issue and must be established before PDT application. Gadolinium (III) and europium (III) oxides either individually or associated with other REEs were preferred because of their validated and well-known application for bioimaging due to their high emission lifetime values [33,34,35]. Additionally, Gd_2_O_3_ incorporated into silica does not dissociate and release into biological tissues; therefore, it is less toxic than Gd chelates, according to Cacheris et al. [36]. However, the genotoxicity of Eu_2_O_3_ without surface functionalization was proven to be considerable for bacterial cell lines, seeds, and algae [37]. D-glucuronic acid-coated Eu_2_O_3_ has been reported to be nontoxic in two parallel tests on cancerous and normal cell lines with an Eu loading of up to 0.5 mmol L^−1^ [38]. In our work, cytotoxicity tests were performed with both Gd_2_O_3_:Eu^3+^ and SiO_2_-Gd_2_O_3_:Eu^3+^ on the mouse glioblastoma cell line GL261 and monkey kidney cells (VERO cell line) using the 3-(4,5-dimethylthiazol-2-yl)-2,5-diphenyltetrazolium bromide (MTT) reduction method.

In this context, the present study comprises physicochemical characterizations and cytotoxicity assays of the synthesized nanoparticles that may be applied in cancer PDT.

## 2. Materials and Methods

### 2.1. Reagents

Europium and gadolinium oxides (99.99%) were purchased from Cstarm Advanced Materials Co., Ltd. (Shanghai, China) and used for the preparation of RE salts. Fumed nanosilica SiO_2_ was provided by Sigma Aldrich (St. Louis, MO, USA), and polyvinyl alcohol (PVA) was provided by ISOFAR.

### 2.2. Synthesis

#### 2.2.1. Synthesis of Rare Earth Salts

Briefly, 0.2 g of Ln_2_O_3_ (Ln = Gd, Eu) was dissolved in 1.0 mL of water. Then, a solution of 99.7% CH_3_COOH was slowly added to form a suspension. The reaction was stirred at 90 °C until the solution became clear. Next, the solvent was evaporated at 60 °C, and dry acetate salt Ln(OAc)_3_·6H_2_O was obtained as a white solid.

#### 2.2.2. Synthesis of Gadolinium Oxide Doped with Europium Nanoparticles

This synthesis was based on the methodology described by Sobral et al. [39] through a sol–gel route. First, Gd(CH_3_COO)_3_·6H_2_O (0.2 g) was mixed with 0.002 g of Eu(CH_3_COO)_3_·6H_2_O, and 2.0 mL of water was added. Then, 1.0 mL of 10% (*w*/*v*) PVA (capping agent) solution was added and homogenized at room temperature for 30 min. The solution was treated at 100 °C for 24 h and then at 200 °C for 5 h. As the last step, the sample was calcined in a muffle furnace at different temperatures (500, 750, and 1000 °C) and times (3 h and 5 h) in an air atmosphere, resulting in a white powder of Gd_2_O_3_:Eu^3+^ with dopant concentrations (Eu^3+^) of 1 wt% concerning the oxide.

According to the infrared (IR) spectra, only nanoparticles treated at 1000 °C for 5 h presented absorption bands related exclusively to Gd-O vibrations. From this, all other characterizations were made only for these samples.

#### 2.2.3. Synthesis of Europium-Doped Gadolinium Oxide Deposited upon Nanosilica

An aqueous solution of Eu(CH_3_COO)_3_ (0.7 mg) was mixed with 256 mg of SiO_2_ and kept in an oven at 100 °C for approximately 1 h to dry. Subsequently, a solution containing 70 mg of Gd(CH_3_COO)_3_∙6H_2_O was gradually added to the previously obtained Eu^3+^-nanosilica. This material was heated in a muffle furnace at 200 °C (1 h), 400 °C (1 h), and 500 °C (1 h). Finally, the hybrid materials labeled SiO_2_-Gd_2_O_3_:Eu^3+^ were calcined at the maximal temperature of 600 °C for 3 h. Maintaining the dopant concentrations (Eu^3+^) for Gd_2_O_3_ (1, 3, and 5 wt%), the samples named SiO_2_-Gd_2_O_3_:Eu^3+^(1%), SiO_2_-Gd_2_O_3_:Eu^3+^(3%), and SiO_2_-Gd_2_O_3_:Eu^3+^(5%) were synthesized as white powders.

### 2.3. Physicochemical Characterization

The IR spectra were recorded in attenuated total reflection (ATR) mode through a Spectrum Two FT-IR spectrometer (Perkin-Elmer, Waltham, MA, USA). X-ray diffraction (XRD) patterns were recorded with an X’Pert PRO X-ray diffractometer (Philips, Toa Payoh, Singapore, PANalytical) using a Cu-K_α_ radiation source (*λ* = 0.154 nm). The red light emission was obtained through X-ray excitation by a diffractometer (Empyrean, PANalytical) using a Cu-K_α_ radiation source (approximately 8 KeV).

The morphology was characterized through scanning electron microscopy (SEM) by employing a JEOL JSM 7100F scanning electron microscope equipped with a silicon drift detector (SDD) for elemental analysis through X-ray energy dispersion spectroscopy (EDS). A multipurpose JEOL JEM 2100F instrument was used to perform transmission electron microscopy (TEM) in conventional (CTEM), high resolution (HRTEM), and scanning (STEM) modes to establish the particle size distribution over the nanosilica matrix.

Photoluminescence analysis was performed using a QuantaMaster 40 (QM 40) UV–Vis spectrofluorometer (Photon Technology International, Pemberton Township, NJ, USA, PTI) with a Xe lamp as an excitation source. Photoluminescence decay data were obtained on a Fluorolog-3 spectrofluorometer (Horiba FL3-22-iHR320, Horiba, Kyoto, Japan) exploiting two dual grids: 1200 g/mm, 330 nm blaze and 1200 g/mm, 500 nm blaze on the excitation and emission monochromators, respectively. The laser with a λ = 980 nm and variable potency (Crystalaser DL980-1WT0, Crystalaser, Reno, NV, USA) served as an excitation source.

DLS and ζ-potential were measured by using an SZ-100 (Horiba) instrument. DLS was performed with an angle of 90° and a duration of 120 s. All measurements were made at room temperature and in triplicate.

### 2.4. Cytotoxic Activity Study

#### 2.4.1. Cell Culture

A mouse glioblastoma cell line (GL261 cells) and a monkey kidney cell line (VERO) were seeded in Dulbecco’s modified Eagle’s medium (DMEM) supplemented with 10% fetal bovine serum (FBS), penicillin (100 U/mL), and streptomycin (100 µg/mL). The cells were incubated at 37 °C (5% CO_2_) following ATCC recommendations. For the experiments, cells (1 × 10^4^ cells/well) were previously seeded in 96-well plates (Greiner Bio-One, Frickenhausen, Germany) and kept for 48 h before each assay (37 °C, 5% CO_2_).

#### 2.4.2. Assessment of the Intrinsic Cytotoxicity of Nanoparticles by the MTT Method

For the cytotoxicity evaluation, cells were seeded as described above and then incubated (37 °C, 5% CO_2_) for 24 h in the presence of different concentrations (10, 31, 62, 125, 250, and 500 µg/mL) of Gd_2_O_3_:Eu^3+^ or SiO_2_-Gd_2_O_3_:Eu^3+^(1%). Next, the cells were washed three times with PBS to remove any nanoparticle residues. The medium was removed; 100 µL of medium with 12.5% (*w*/*v*) MTT (5 mg/mL) and without phenol red were added to each well; and the cells were incubated for 3 h in the dark. After incubation, the medium was removed, and 100 µL of DMSO were added to solubilize the formed formazan crystals. The absorbance was measured at *λ* = 570 nm with a Spectramax 190 spectrophotometer. The experiments in GL261 were performed in triplicate on four independent days for Gd_2_O_3_:Eu^3+^ and three independent days for SiO_2_-Gd_2_O_3_:Eu^3+^(1%). The experiments in Vero were performed in triplicate on three independent days for both nanoparticles. Untreated cells were used as a negative control, and 1% (*v*/*v*) Triton X-100 or 5% (*v*/*v*) Tween 20 was added to cells as a control for cell death. The cell viability was calculated using the mean absorbance of controls (cells without any treatment), which was considered 100% viability. Then, the following formula was used to obtain the percentage of viability for each condition tested:(1)$$Viability\;\left(\%\right)=\frac{ABS\;Treatment}{ABS\;Controls}\times 100$$

#### 2.4.3. Statistical Analysis

Data homoscedasticity was verified with Bartlett’s test, and normality was checked with a D’Agostino–Pearson test. When the data were homoscedastic and normally distributed, an ANOVA with a Bonferroni posttest was used for multiple comparisons. For nonnormal distributions, the Kruskal–Wallis test followed by Dunn’s posttest was applied. The statistical analysis was performed using the GraphPad Prism 7 program (San Diego, CA, USA). All data were expressed in a graph by the arithmetic mean ± standard error of the mean, and the criterion at *p* < α (α = 0.05) was considered to be significant.

## 3. Results and Discussion

### 3.1. Infrared Spectroscopy

The gadolinium acetate salt used as a precursor in the synthesis of nanoparticles was studied first by using infrared spectroscopy (IR spectroscopy). The IR spectrum (Figure 1a) illustrates two absorbance bands at 3357 and 3246 cm^−1^, probably from free and complex water molecules in the structure [40,41]. The band at 1545 cm^−1^ is assigned to νas(COO-), the 1455 cm^−1^ band to νs(COO-), the 1414 cm-1 band to δas(CH_3_), and the 1354 cm^−1^ band to δs(CH_3_). The bands in the regions of 675 cm^−1^ and 610 cm^−1^ were associated with δ(O-C-O). Lastly, the vibration band of the Gd-O bond appears at 439 cm^−1^. In the Gd_2_O_3_:Eu^3+^ nanoparticles, this band appears at 542 and 436 cm^−1^ [40,42,43,44]. No bands referring to organic intermediates were detected, and no O-H bands were found, indicating the absence of water molecules in these nanoparticles (Figure 1b).

For comparison purposes, Gd_2_O_3_:Eu^3+^ nanoparticles calcined at different temperatures and durations were also studied with IR spectroscopy (Figure 2). The band corresponding to the water molecules is observed at 3410 cm^−1^. Absorption bands associated with carbonate anions formed during the calcination process within the region of 1500–1390 cm^−1^ and at 847 cm^−1^ are observed [40]. Two bands near 3000 cm^−1^ are characteristic bands for C-H, or more specifically, from the stretching of CH_2_ groups and asymmetric stretching of CH_3_ groups, confirming the presence of organic groups [43,45,46]. Therefore, the decrease in the intensity of all these bands and their consequent disappearances as the calcination temperature and time increased leads to the conclusion that water and organic materials were eliminated during thermal treatment at 1000 °C for 5 h.

In addition to the characteristic bands of Ln-O in the IR spectrum of SiO_2_-Gd_2_O_3_:Eu^3+^(1%) nanoparticles, two bands referring to SiO_2_ are evident (Figure 3); the intense band at 1080 cm^−1^ is assigned to the asymmetric stretching vibration of Si-O-Si, whereas the band at 807 cm^−1^ is attributed to the symmetrical stretching vibration of Si-O-Si [46,47,48]. The low intensity and wide band width at 3355 cm^−1^ are typical for OH^−^, which is responsible for moisture inside the sample [41,43,49]. The characteristic band of Gd-O appears to be at 450 cm^−1^, probably being overlapped by the bending vibration mode of Si-O, which also appears in this region [47].

### 3.2. X-ray Diffraction

The X-ray diffraction (XRD) profile for Gd_2_O_3_:Eu^3+^ shows typical diffraction peaks for the cubic crystalline Gd_2_O_3_ phase (d_Scherrer_ = 32 nm), as shown in Figure 4. On the other hand, SiO_2_-Gd_2_O_3_:Eu^3+^(1%) demonstrates a strong amorphous background with a negligible reflex of crystalline Gd_2_O_3_ (Figure 4). The amorphous signal is obviously due to the amorphous nanosilica support, and the low intensity of the reflections from SiO_2_-Gd_2_O_3_:Eu^3+^(1%) may be due to the low RE/SiO_2_ mass ratio.

As seen in the standard spectra (Figure 4), the diffractograms for the Gd^3+^ and Eu^3+^ oxides are hardly distinguishable due to their identical cubic space groups and lattice configurations and their similar lattice parameters (1.0808 nm for Gd_2_O_3_ and 1.0866 nm for Eu_2_O_3_), which is quite expected. However, the XRD results confirmed the formation of the crystalline phase of Gd_2_O_3_:Eu^3+^.

### 3.3. Electron Microscopy

#### 3.3.1. Scanning Electron Microscopy

The scanning electron microscopy (SEM) images of Gd_2_O_3_:Eu^3+^ show aggregates of uniformly distributed primary nanoparticles possessing an isotropic shape with a mean size of approximately 100 nm (Figure 5). Silica-supported Gd_2_O_3_:Eu^3+^ nanoparticles show a smaller size and homogeneous distribution with an agglomeration of these nanoparticles, as can be seen in Figure 6a,b. We show SEM images of one concentration, SiO_2_-Gd_2_O_3_:Eu^3+^(1%), and the results of the other two samples (at 3 and 5% Eu^3+^) are in the Appendix A.

The elements in SiO_2_-Gd_2_O_3_:Eu^3+^(1%) nanoparticles were also mapped by X-ray energy dispersion spectroscopy (EDS) elemental mapping (Figure 7) to detect the Gd and Eu distributed congruently on silica, despite the low intensities of their signals. Notably, O and Si are the most visible elements. These elements are homogeneously distributed in the nanosilica matrix.

It is difficult to determine the precise size of the two samples with only SEM results. We studied silica-free Gd_2_O_3_:Eu^3+^ and SiO_2_-Gd_2_O_3_:Eu^3+^(1%) using transmission electron microscopy (TEM) and transmission electron microscopy in high-resolution (HRTEM) mode to establish precise size distributions of the active phases—RE oxides.

#### 3.3.2. Transmission Electron Microscopy

Regarding Gd_2_O_3_:Eu^3+^, the TEM images of the nanoparticles demonstrate that despite being aggregated, they have a homogeneous distribution with an average diameter of ~100 nm (Figure 8a,b), which is consistent with the SEM results shown above. The clear grain boundaries inside the aggregates indicate a distinct delimitation between the Gd_2_O_3_ crystallites (Figure 8c,d). Additionally, the EDS results confirm the presence of Eu and Gd in the sample, and the elemental mapping results also reveal that Eu is homogeneously incorporated into the Gd_2_O_3_ nanoparticles (Figure 9).

Regarding SiO_2_-Gd_2_O_3_:Eu^3+^(1%) nanoparticles, one can discern the amorphous phase of silica at a larger particle size, and the rare earth metal oxide is essentially smaller than the respective support (Figure 10a,b). The crystallinity of the rare metal oxide and overall morphology of the sample are shown by different TEM and scanning transmission electron microscopy (STEM) images in the Appendix A. To support this observation, bright-field and high-angle annular dark-field images (HAADF) in STEM mode were obtained (Figure 10c,d). The HAADF image has atomic number contrast and hence clearly shows that the brighter oxide nanoparticles in Figure 10d contain much heavier atoms (Gd and Eu) than the matrix.

The size distribution of gadolinium oxide nanoparticles over nanosilica was assessed and plotted through a histogram with a basic statistical analysis (Figure 11). From this figure, we can observe that most of the nanoparticles are in the range of 2–7 nm, which is much smaller than the Gd_2_O_3_:Eu^3+^ sample.

### 3.4. Photoluminescence Studies

The room temperature excitation and emission spectra (UV–Vis) of the Gd_2_O_3_:Eu^3+^ nanoparticles were recorded, and the results confirmed the characteristic transitions of Eu^3+^ ions (Figure 12).

The excitation spectrum (Figure 12a) demonstrates the typical bands of this material. The broad absorption band at 260 nm refers to the ligand-to-metal charge transfer (LMCT) transition from O^2−^ (2p) to Eu^3+^ (4f^6^) ions. The ^8^S_7/2_ → ^6^I_7/2,9/2,17/2_ transitions characteristic of Gd^3+^ ions overlap with LMCT and appear as a shoulder at approximately 280 nm. The band at 313 nm is due to electronic transitions (^8^S_7/2_ → ^6^P_7/2,5/2,3/2_) of the Gd^3+^ ions, which involve energy transfers from Gd^3+^ to Eu^3+^. The absorption bands at 467 nm and 536 nm are attributed to the ^7^F_0_ → ^5^D_2_ and ^7^F_0,1_ → ^5^D_1_ transitions of the Eu^3+^ ion [41].

In the emission spectrum, well-defined characteristic emission bands of Eu^3+^ are observed (Figure 12b). All transitions start from the ^5^D_0_ state since other excited state transitions, such as ^5^D_1_, ^5^D_2_, and ^5^D_3_, are much less common. These characteristic bands refer to the ^5^D_0_ → ^7^F_J_ transitions (J = 0 to 4) of Eu^3+^; more specifically, to the ^5^D_0_ → ^7^F_0_ (580 nm), ^5^D_0_ → ^7^F_1_ (591 nm), ^5^D_0_ → ^7^F_2_ (610 nm), ^5^D_0_ → ^7^F_3_ (649 nm), and ^5^D_0_ → ^7^F_4_ (707 nm) transitions. The ^5^D_0_ → ^7^F_5_ and ^5^D_0_ → ^7^F_6_ transitions are not observed since they appear in the IR region and the intensities of these transitions are very low [23,50].

The photoluminescence emission spectra of SiO_2_-Gd_2_O_3_:Eu^3+^(1%) nanoparticles were recorded in a spectral range from 550 nm to 720 nm, with excitation at a wavelength of 260 nm (Figure 13).

All the profiles of SiO_2_-Gd_2_O_3_:Eu^3+^ nanoparticles present one unique intense band typical for trivalent europium ions at 610 nm, similar to Gd_2_O_3_:Eu^3+^ (Figure 12b). Comparing the spectra, SiO_2_-Gd_2_O_3_:Eu^3+^(3%) shows the highest emission intensity. When the Eu^3+^ concentration is further increased to 5%, the emission intensity decreases but is still stronger than that of SiO_2_-Gd_2_O_3_:Eu^3+^(1%). The suppression of luminescence induced by concentration is quite common for doped materials.

Since the main objective of our work is the future use of these nanoparticles in the treatment of deep tumors, we performed a luminescence study for Gd_2_O_3_:Eu^3+^ nanoparticles using an X-ray source, and it was possible to observe the scintillating effects of these nanoparticles through the emission of red light in the visible region. The image obtained is presented in the Appendix A.

The photoluminescence decay was measured (Figure 14) to calculate the lifetime of the Gd_2_O_3_:Eu^3+^ nanoparticles. The lifetime of the excited Eu^3+^ level in the analyzed nanoparticles was *t_1_* = 1.31 ms, which is similar to the reported values in the literature [41]. The first-order exponential decay proves the homogeneity and purity of the material.

### 3.5. Stability Measures

#### 3.5.1. Dynamic Light Scattering (DLS)

The hydrodynamic size and the distribution of the nanoparticles synthesized were determined through DLS measurements (Figure 15 and Figure 16).

It can be seen that Gd_2_O_3_:Eu^3+^ nanoparticles showed two different size distributions (Figure 15). The first peak indicates an average hydrodynamic radius of 81.3 nm and the second, an average of 333.7 nm. From this result, it seems that these nanoparticles aggregate in an aqueous solution forming an agglomerate, which is in agreement with the results obtained in electron microscopy. The first peak probably corresponds to the real size of the nanoparticles. This would justify the variation in the size of the hydrodynamic radius in the second peak since these NPs can aggregate to form clusters of different sizes.

Two different size distributions were also observed in SiO_2_-Gd_2_O_3_:Eu^3+^ nanoparticles (Figure 16): the first peak had an average hydrodynamic radius value of 65.7 nm and the second had an average of 365.6 nm. These nanoparticles also appeared to be aggregating in an aqueous solution; the second peak varied more than the first and was distributed with a lower frequency.

Therefore, these nanoparticles are not very stable in an aqueous solution as they tend to aggregate, which is not ideal. Thus, the stability of these nanoparticles needs to be improved.

The autocorrelation functions referring to DLS measurements were obtained as depicted in the Appendix A.

#### 3.5.2. ζ-Potential

The ζ-potentials of the nanoparticle suspensions are stable negative values of −24.9 mV for Gd_2_O_3_:Eu^3+^ and −31.9 mV for SiO_2_-Gd_2_O_3_:Eu^3+^(1%) (Figure 17). A slight increase in the charge in the latter sample is due to the presence of excess silica, and its ζ-potential is over the well-known critical value of −30 mV, which is a stability criterion for pure electrostatic repulsion between nanoparticles. For this reason, we cannot speculate on the pure contribution of electrostatic interaction in the subject suspensions because the adsorption of water molecules at the active sites on the oxide surface occurs easily, especially in the case of nanosilica. Considering the actual ζ-potential for SiO_2_-Gd_2_O_3_:Eu^3+^(1%), superficially active –OH groups, and the size from the SEM images, we may expect this sample to demonstrate higher stability in normal saline solutions utilized for biomedical applications.

### 3.6. Cytotoxic Activity Study

To evaluate the intrinsic cytotoxicity of the Gd_2_O_3_:Eu^3+^ and SiO_2_-Gd_2_O_3_:Eu^3+^(1%) nanoparticles, the GL261 cell line was treated with different concentrations (10, 31, 62, 125, 250, and 500 µg/mL) of the nanoparticles diluted in culture medium, and the cell viability was evaluated by the MTT assay (Figure 18). ANOVA with a Bonferroni posttest and a Kruskal–Wallis test with a Dunn posttest showed that there is no significant difference between the negative control and treatment with different concentrations of Gd_2_O_3_:Eu^3+^ (Figure 18a) and SiO_2_-Gd_2_O_3_:Eu^3+^(1%) (Figure 18b) nanoparticles. However, when testing the same nanoparticles in non-tumor cell lines (Figure 19), we observed a small but statistically significant reduction in viability with 500 µg/mL of Gd_2_O_3_:Eu^3+^ (Figure 19a) and with 62 and 500 µg/mL of SiO_2_-Gd_2_O_3_:Eu^3+^(1%) (Figure 19b).

Gadolinium nanoparticles have been used in the development of new markers for MRI [51,52] and have relatively low cytotoxicity and immunotoxicity evaluated in vivo [26,53]. Gd2O3:Eu3+ nanoparticles also show few negative systemic effects, indicating the good biocompatibility of these NPs [54]. During the cytotoxicity assays, we observed some difficulty in keeping a stable nanoparticle suspension. This fact may have influenced our results, with some variation in the local concentration of nanoparticles available to interact with the cell lines. To avoid this issue, it may be interesting to work with fresh nanoparticle solutions prepared daily. The use of an ultrasonic pulse with increased temperature can help to reduce this aggregation.

Nanoparticles have been regarded mainly as passive carriers in PDT, but some formulations have been described whereby carrier nanoparticles have an additional active role in the process of photodynamic activation [10,55,56]. Besides, an important aspect to be addressed when applying molecules or nanoparticles in clinical treatments is their intrinsic toxicity because when preparing the nanoparticle–photosensitizer conjugates, it is important to know the cytotoxicity generated by each one. The nanoparticles tested in this study, as shown in Figure 18 and Figure 19, presented small cytotoxicity toward both GL261 and VERO cells. Therefore, the Gd_2_O_3_:Eu^3+^ and SiO_2_-Gd_2_O_3_:Eu^3+^(1%) nanoparticles are good candidates to be tested in vitro photodynamic therapy assays on nanoplatforms with well-known photosensitizers such as methylene blue.

## 4. Conclusions

Gd_2_O_3_:Eu^3+^ and SiO_2_-Gd_2_O_3_:Eu^3+^ nanoparticles were prepared by the sol–gel synthesis and impregnation methods, respectively. The crystalline phase of the rare earth oxide (Gd_2_O_3_:Eu^3+^) was distributed over the entire nanosilica matrix in SiO_2_-Gd_2_O_3_:Eu^3+^, whose ζ-potential is more negative than that of silica-free Gd_2_O_3_:Eu^3+^. Nanosilica reduced the particle size of the rare earth oxides to an average diameter of 4.9 nm, in contrast to pure Gd_2_O_3_:Eu^3+^ nanoparticles (100 nm). The nonlinear intensity of photoluminescence as a function of Eu^3+^ concentration manifests a maximum at 3% Eu^3+^ in SiO_2_-Gd_2_O_3_:Eu^3+^, in which the emission spectrum exhibits the intense ^5^D_0_ → ^7^F_2_ transition of Eu^3+^, as well as in Gd_2_O_3_:Eu^3+^. Light emission in the visible region was also observed for Gd_2_O_3_:Eu^3+^ nanoparticles when excited by X-rays. The size and surface charge of the studied samples reveal their suitability for applications in biological systems by successfully passing cytotoxicity tests in cancer cell lines and non-tumor cell lines. These experimental results demonstrate the potential of rare earth-based luminescent nanoparticles as a promising alternative in photodynamic therapy.

## Figures and Tables

**Figure 1 pharmaceutics-14-02258-f001:**
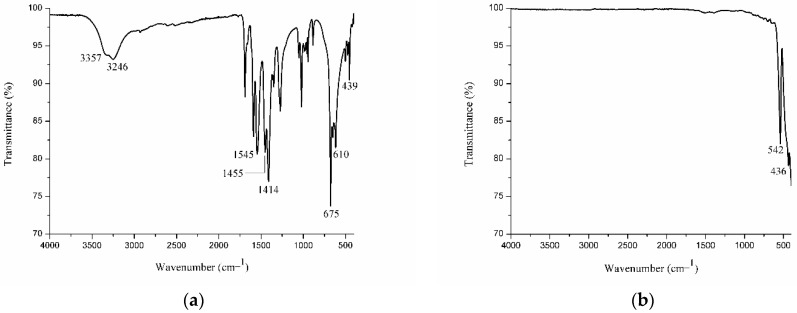
IR spectrum showing the characteristic absorption bands of the (**a**) precursor Gd(CH_3_COO)_3_·6H_2_O salt and (**b**) Gd_2_O3:Eu^3+^ nanoparticles synthesized from the salt and calcined at 1000 °C for 5 h.

**Figure 2 pharmaceutics-14-02258-f002:**
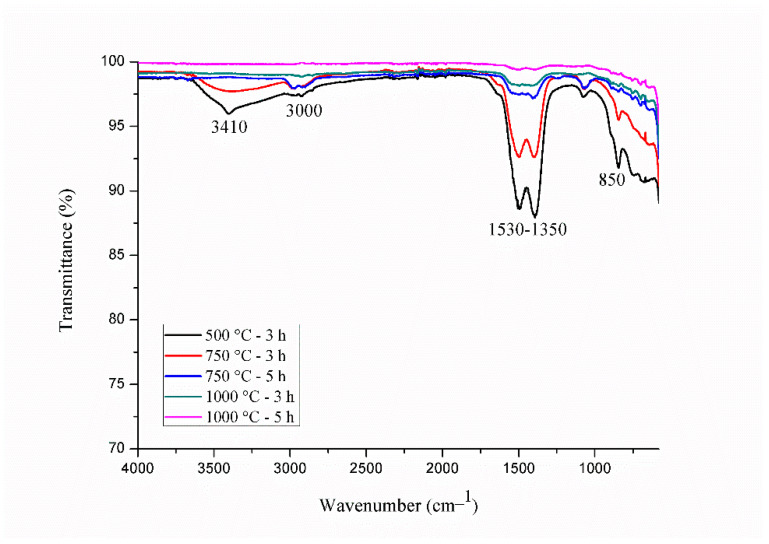
IR spectra of Gd_2_O_3_:Eu^3+^ nanoparticles calcined at different temperatures and times.

**Figure 3 pharmaceutics-14-02258-f003:**
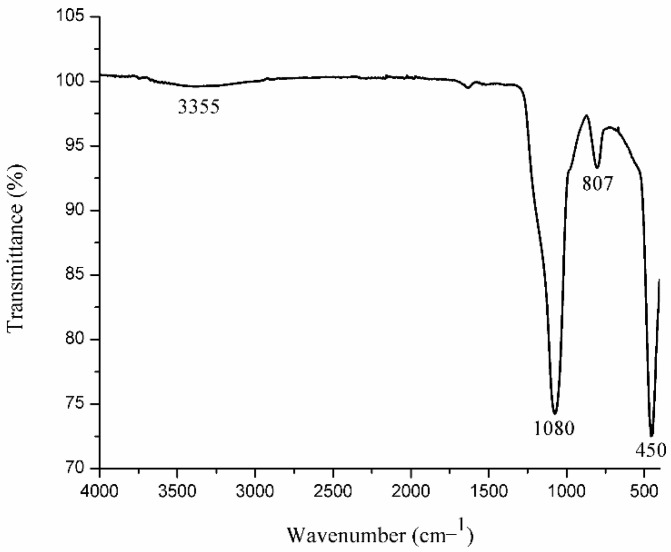
IR spectrum of the SiO_2_-Gd_2_O3:Eu^3+^(1%) nanoparticles, displaying the typical absorption bands of this material.

**Figure 4 pharmaceutics-14-02258-f004:**
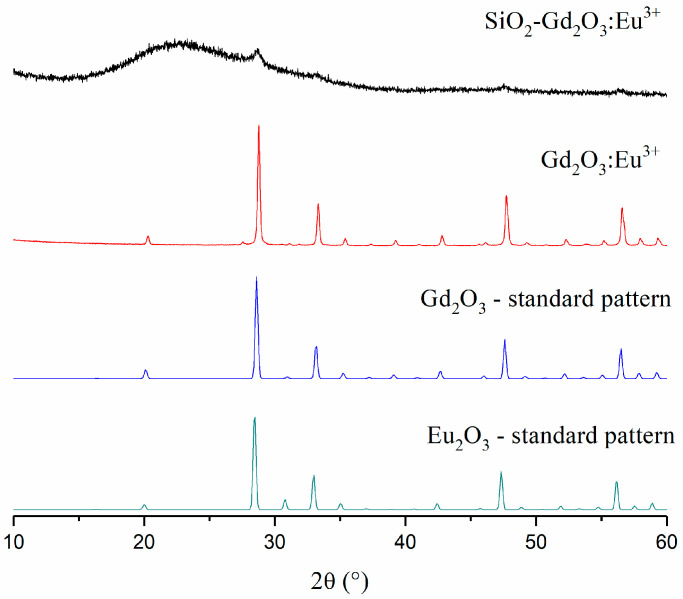
XRD patterns of SiO_2_-Gd_2_O_3_:Eu^3+^(1%), Gd_2_O_3_:Eu^3+^, and crystalline standard phases: Gd_2_O_3_ (ICSD#184590) and Eu_2_O_3_ (ICSD#194513).

**Figure 5 pharmaceutics-14-02258-f005:**
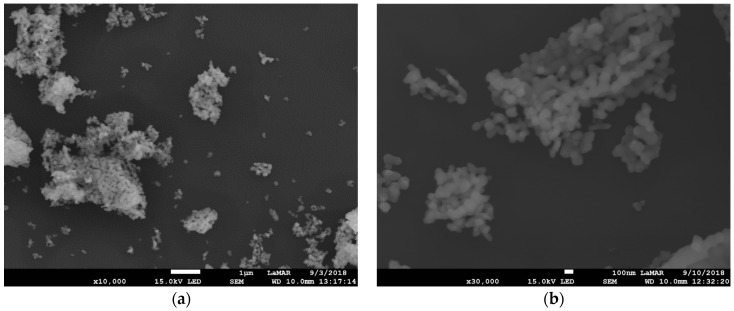
SEM image of Gd_2_O_3_:Eu^3+^ nanoparticles at 10^4^ times magnification (**a**) showing the aggregates of uniformly distributed nanoparticles and SEM image at 3 × 10^4^ augmentation and (**b**) showing that the nanoparticles present an isotropic shape with a mean size of approximately 100 nm. The scale bar in (**a**) corresponds to 1 μm and in (**b**) to 100 nm.

**Figure 6 pharmaceutics-14-02258-f006:**
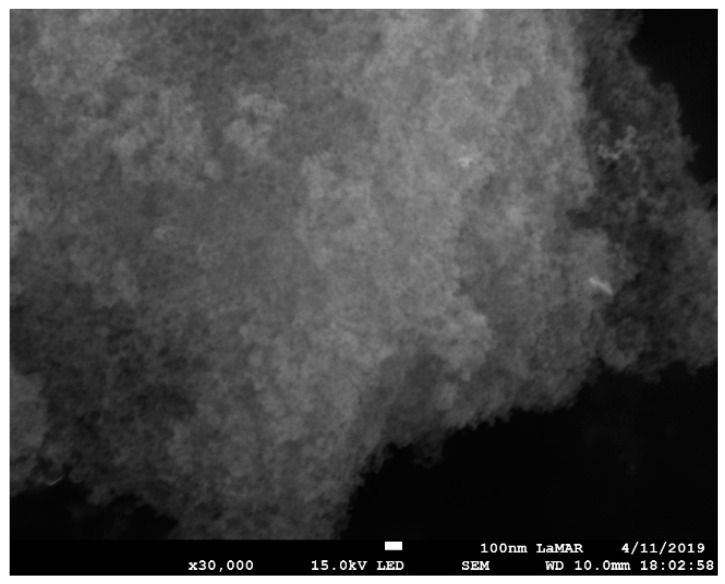
SEM image of SiO_2_-Gd_2_O_3_:Eu^3+^(1%) nanoparticles at 3 × 10^4^ times magnification, revealing aggregates with homogeneous distribution. The mean size of the silica nanoparticles is much smaller than that of the silica-free nanoparticles. The scale bar corresponds to 100 nm.

**Figure 7 pharmaceutics-14-02258-f007:**
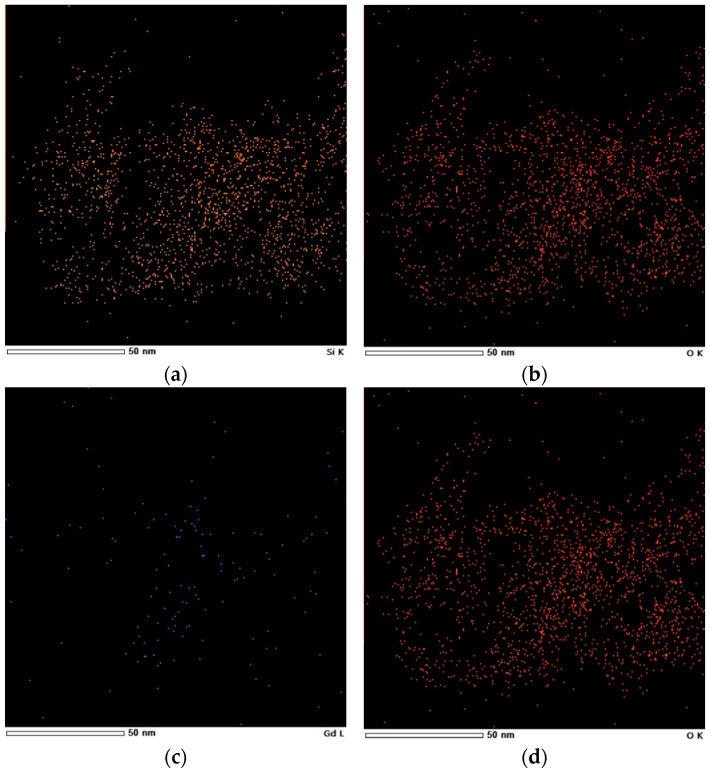
Elemental mapping of (**a**) Si, (**b**) O, (**c**) Gd, and (**d**) Eu in SiO_2_-Gd_2_O_3_:Eu^3+^(1%), showing the distribution of these elements in the nanoparticles.

**Figure 8 pharmaceutics-14-02258-f008:**
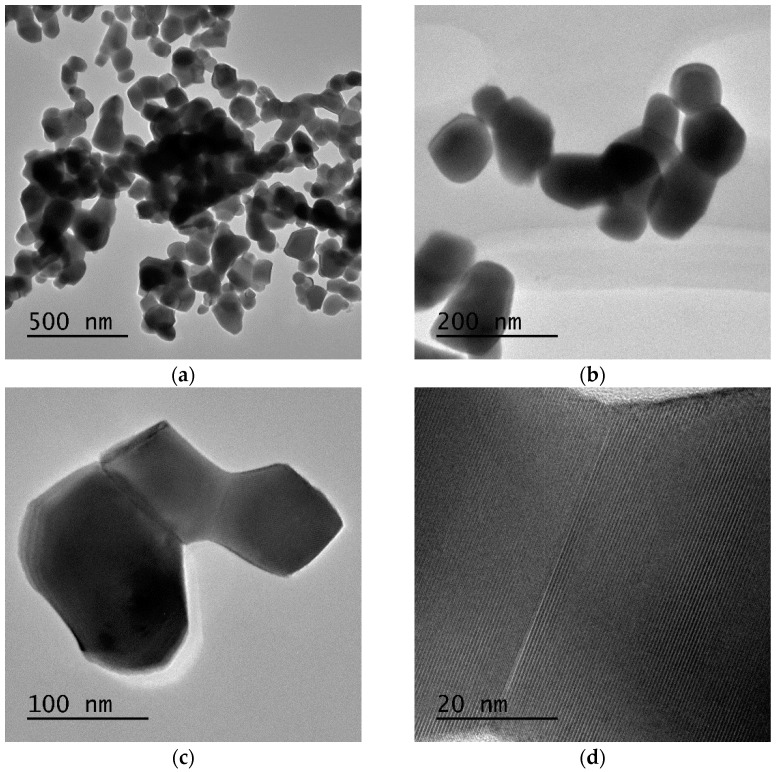
(**a**,**b**) Gd_2_O_3_:Eu^3+^ nanoparticles have a homogeneous character with an average diameter of approximately 100 nm. (**c**,**d**) Distinct grain boundaries inside the aggregates indicate delimitation between the Gd_2_O_3_ crystallites.

**Figure 9 pharmaceutics-14-02258-f009:**
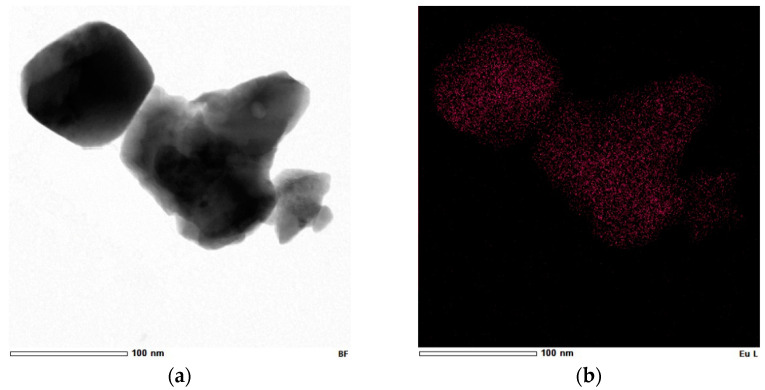
(**a**) Bright-field STEM image and elemental mapping confirming the presence of (**b**) Eu, (**c**) Gd, and (**d**) O in the Gd_2_O_3_:Eu^3+^ nanoparticles.

**Figure 10 pharmaceutics-14-02258-f010:**
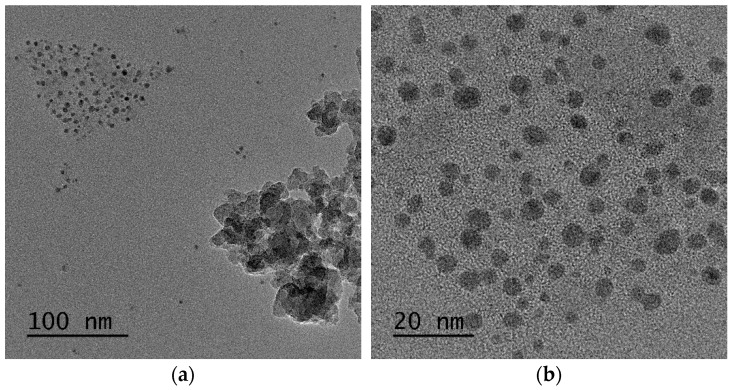
(**a**,**b**) STEM images in which the amorphous phase of silica with larger particle size and smaller rare earth oxide particles are distinguishable. (**c**) Bright-field and (**d**) HAADF images show that the oxide nanoparticles contain much heavier atoms than the matrix.

**Figure 11 pharmaceutics-14-02258-f011:**
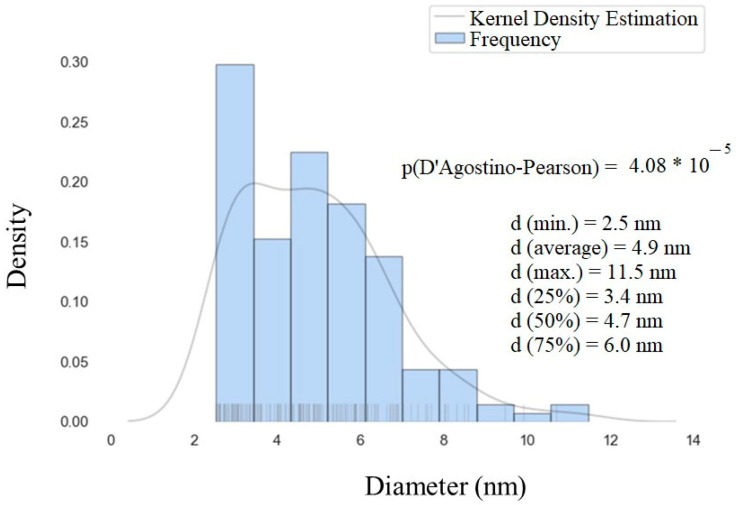
Histogram of the particle size distribution obtained from the STEM images of SiO_2_-Gd_2_O_3_:Eu^3+^(1%).

**Figure 12 pharmaceutics-14-02258-f012:**
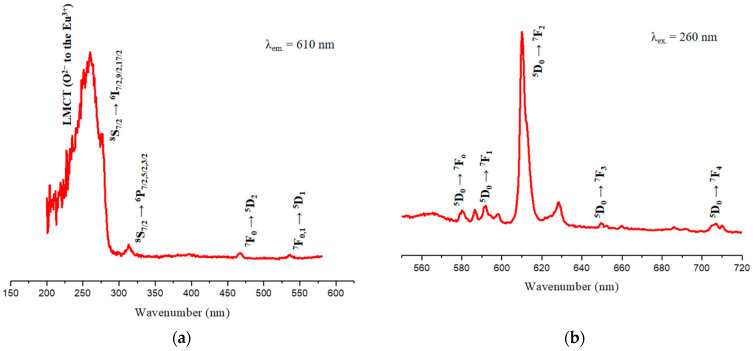
Excitation (**a**) and emission (**b**) spectra of the Gd_2_O_3_:Eu^3+^ nanoparticles, representing the different electronic transitions that occur in this material.

**Figure 13 pharmaceutics-14-02258-f013:**
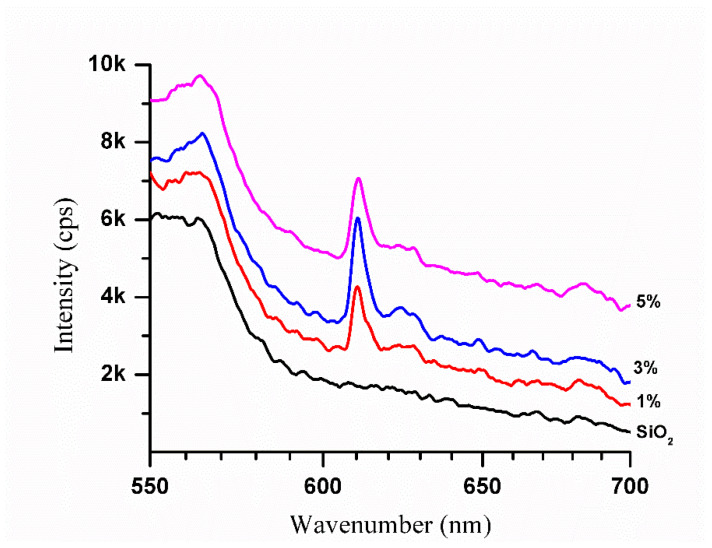
Photoluminescence emission spectra of pure SiO_2_ and SiO_2_-Gd_2_O_3_:Eu^3+^(1%) nanoparticles at different europium ion concentrations (1, 3, and 5%). The spectra exhibit an intense band corresponding to the europium ion.

**Figure 14 pharmaceutics-14-02258-f014:**
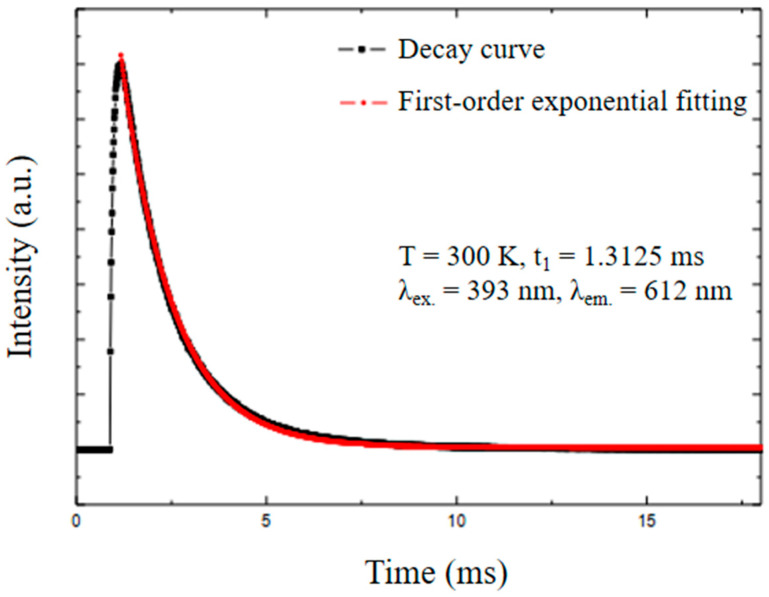
Photoluminescence decay of the Gd_2_O_3_:Eu^3+^ nanoparticles, indicating the emission lifetime of this material.

**Figure 15 pharmaceutics-14-02258-f015:**
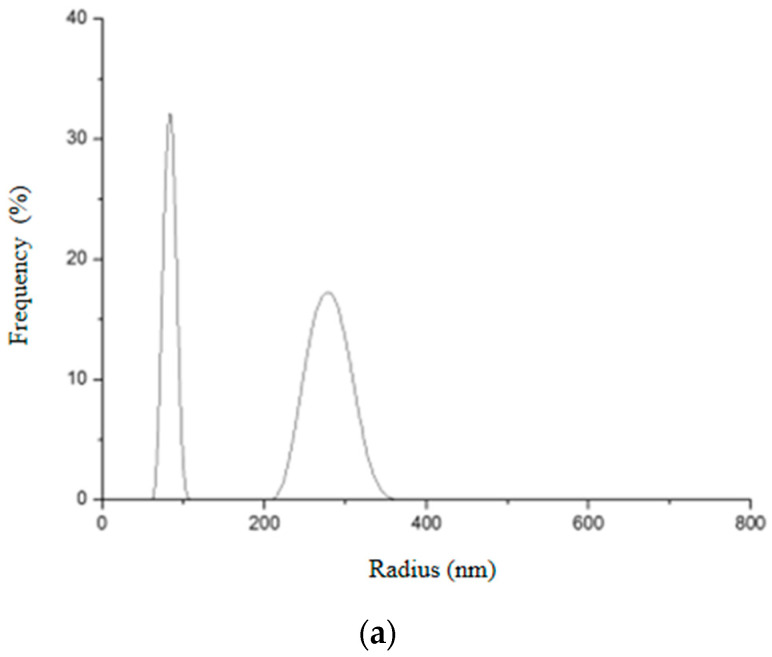
Hydrodynamic radius distribution measured in triplicate (**a**–**c**) for Gd_2_O_3_:Eu^3+^ nanoparticles, indicating that they tend to form clusters of different sizes in aqueous solution.

**Figure 16 pharmaceutics-14-02258-f016:**
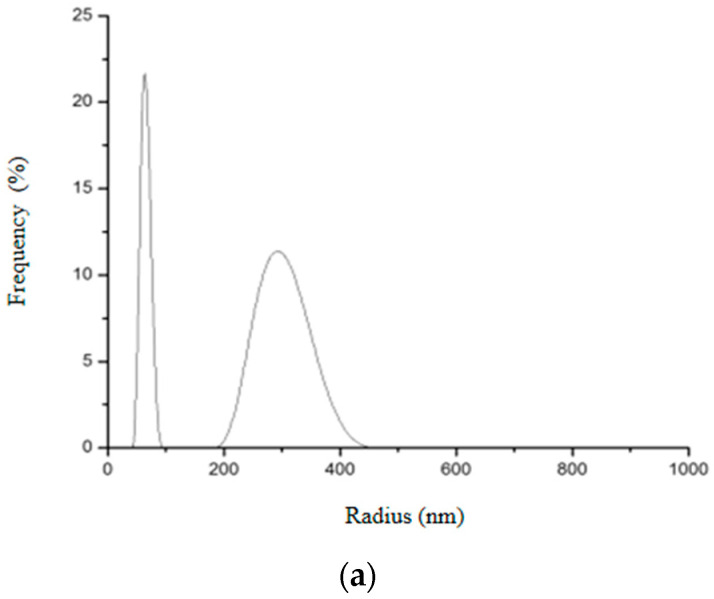
Hydrodynamic radius distribution measured in triplicate (**a**–**c**) for SiO_2_-Gd_2_O_3_:Eu^3+^ nanoparticles, indicating that they also tend to form clusters of different sizes in aqueous solution.

**Figure 17 pharmaceutics-14-02258-f017:**
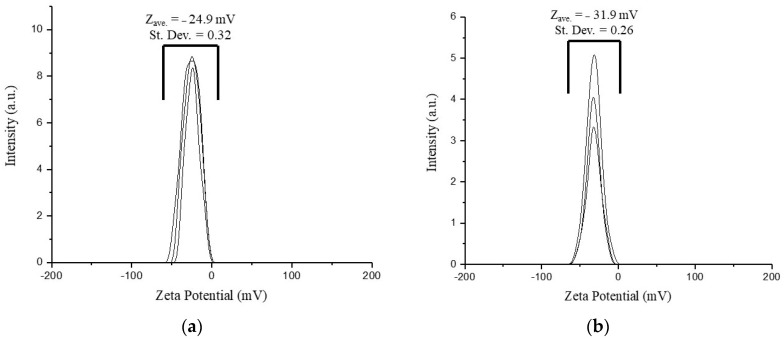
ζ-potential diagram measured in triplicate for the (**a**) Gd_2_O_3_:Eu^3+^ and (**b**) SiO_2_-Gd_2_O_3_:Eu^3+^(1%) nanoparticles, in which negative potential values are found for both samples.

**Figure 18 pharmaceutics-14-02258-f018:**
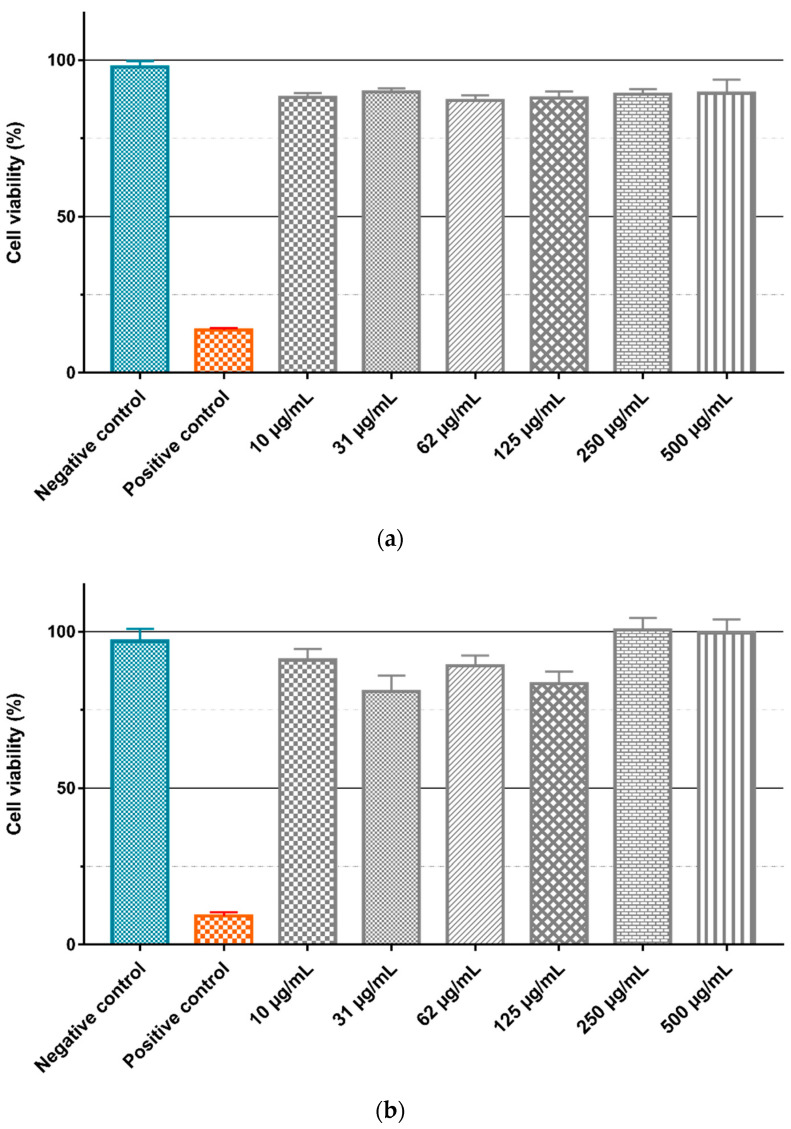
Evaluation of nanoparticle cytotoxicity by the MTT method. The GL261 tumor cell line was treated for 24 h in a suspension containing different concentrations of (**a**) Gd_2_O_3_:Eu^3+^ and (**b**) SiO_2_-Gd_2_O_3_:Eu^3+^(1%) nanoparticles. The graphics represent the mean values + standard deviation of the mean of (**a**) four independent experiments and (**b**) three independent experiments. Negative control: cells in medium only. Positive control: cells treated with (**a**) 0.4% (*v*/*v*) Triton X-100 or (**b**) 0.5% (*v*/*v*) Tween 20.

**Figure 19 pharmaceutics-14-02258-f019:**
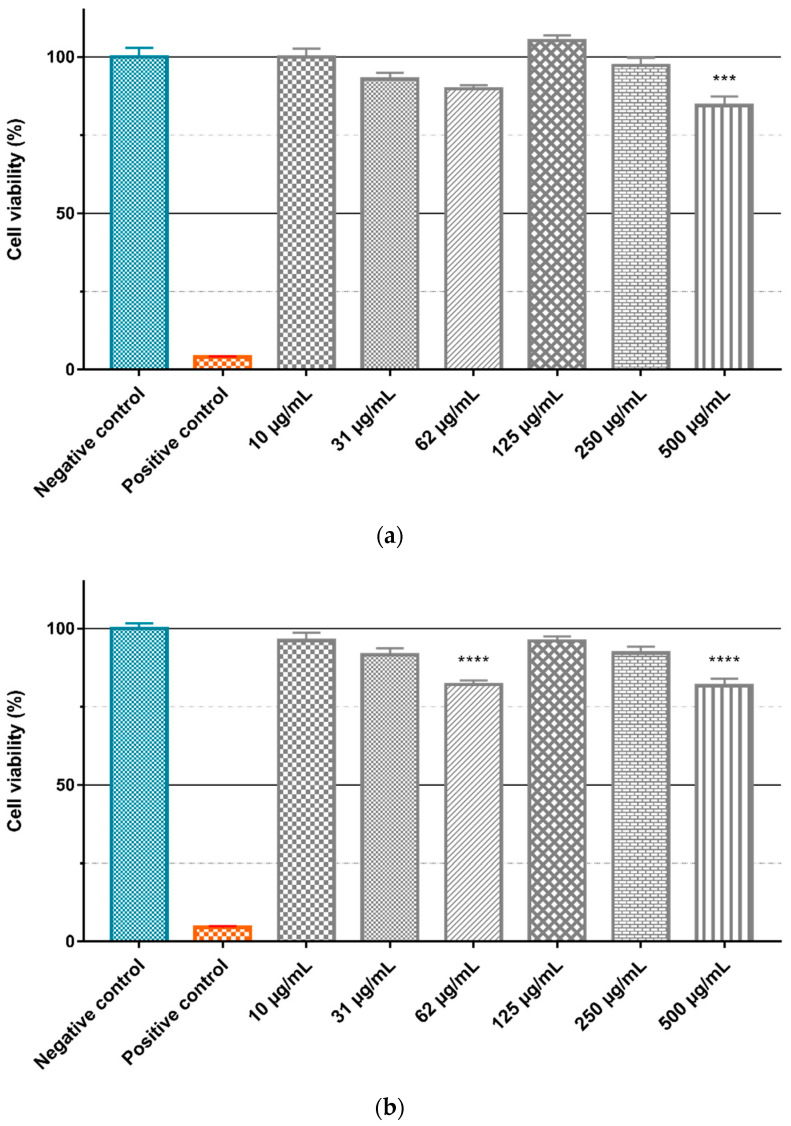
Evaluation of nanoparticle cytotoxicity by the MTT method. The VERO cell line was treated for 24 h in a suspension containing different concentrations of (**a**) Gd_2_O_3_:Eu^3+^ and (**b**) SiO_2_-Gd_2_O_3_:Eu^3+^(1%) nanoparticles. The graphics represent the mean values + standard deviation of the means of three independent experiments. Negative control: cells in medium only. Positive control: cells treated with 0.4% (*v*/*v*) Triton X-100 (*** *p* = 0.0001, **** *p* < 0.0001).

## Data Availability

The data presented in this study are available in the article and its Appendix A.

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
