# Peer review of "Novel Scintillating Nanoparticles for Potential Application in Photodynamic Cancer Therapy"

_pharmaceutics, 2022, doi:10.3390/pharmaceutics14112258_

Round 1

Reviewer 1 Report (Previous Reviewer 1)

In this work, the authors fabricated a hybrid silica nanoparticles evaluated the potential applications for photodynamic tumor therapy. Overall, the materials synthesis of SiO2-Gd2O3:Eu3+ was clear indicated, but the biomedical evaluations were too premature to be published. Some specific comments should be addressed before further consideration:

1. In Figure 5, some aggregates were observed in the SEM images. The authors were suggested to provide a clear images with good dispersibility of SiO2-Gd2O3:Eu3+.

2. Figure 6 was too blurry, which should improve the resolution. Moreover, Figure 6a and 6b should be added in the captions, which should be discussed in the manuscript respectively.

3. The aggregates were also observed in Figure 8, and the figures should also be discussed in the manuscript.

4. The particle size in Figure 10 and 11 was not consistent with the previous results (Figure 5-9)

5. In Figure 12, it was observed that the maximize absorption of SiO2-Gd2O3:Eu3+ was around 260 nm, which was not suitable for in vitro and in vivo applications.

6. The biocompatibility of SiO2-Gd2O3:Eu3+ was evaluated in Figure 18-19, but the PDT potential of SiO2-Gd2O3:Eu3+ was suggested to be investigated.

Author Response

The authors would like to thank the reviewers and the Editor for their time and their valuable comments concerning our manuscript.

As you can see, we have considered all reviewer suggestions. The critiques were addressed, and changes were made to the manuscript. We hope this manuscript will now be considered suitable for publication.

Reviewer 2 Report (New Reviewer)

The paper presents development of X-ray absorbing scintillating gadolinium oxide nanoparticles doped with europium ions. Nanoparticles were developed for their potential use as alternative materials in PDT.  The particles were structurally characterized and tested on biocompatibility. Cytotoxicity was studied in vitro on two cell lines. The paper presents an extended experimental data and is well written. The paper can be considered worth of being published in Pharmaceutics after addressing the following concerns.

-       -  My major concern is on photoluminescence properties of the particles. The paper presents the absorption spectrum in UV-Vis which shows one main band peaked at about 260 nm, within the range where absorption of tissues is even higher than in visible… Since the main stated goal is the development of X-ray absorbing scintillating nanoparticles, the absorption spectrum in the corresponding range is required.  Along with the corresponding emission spectrum. Addition of the absorption spectrum of, say, blood in the same range would be also useful.

-       -  Since the designed nanoparticles are aimed to be used for PDT special attention should be given to their toxicity not only on the cellular level, but also for the entire organism. The ways and rates of their clearance from the body are also very important. The detailed discussion on this subject concerning similar nanoparticles and with references to corresponding investigations is necessary.  

-       -  The ways of particles administration in vivo is also an important point.  Some discussion on that is required also.

-        - The aggregation of particles is a serious problem. Would be rewarding to add some comments on potential ways how to avoid that. Besides, experiments with cells were performed in DMEM. The analysis of particles aggregation in the culture medium and PBS could help to work on the problem.

Some minor concerns

-       -  The statement on pages 3-4 that viability of cancer cells with nanoparticles was tested because the particles were developed for anticancer applications sounds controversial. I would recommend to remove this reason and just mention that viability of cells of the two lines, cancerous and pseudonormal, was tested.

-      -  The caption to Fig. 12 is erroneous. Peaks in (b) are by no means absorption bands.  

-     -   The part of text on page 19 concerning description of PDT should be moved to the Introduction section.

-      -  The manuscript title is not the same as that in the supplementary material.

Author Response

The authors would like to thank the reviewers and the Editor for their time and their valuable comments concerning our manuscript.

As you can see, we have considered all reviewer suggestions. The critiques were addressed and changes were made to the manuscript. We hope this manuscript will now be considered suitable for publication.

Round 2

Reviewer 2 Report (New Reviewer)

I am satisfied with the corrections made by the authors to address my concerns. The paper can be accepted in the current form.

This manuscript is a resubmission of an earlier submission. The following is a list of the peer review reports and author responses from that submission.

Round 1

Reviewer 1 Report

In this work, the authors synthesized and characterized SiO2-Gd2O3:Eu3+. It was confirmed that the nanoparticles displayed stable negative zeta potential and photoluminescence properties. However, the PDT properties of SiO2-Gd2O3:Eu3+ were not investigated, which was not coincident with the topic of this work. Some specific issues were listed as bellow:

  1. The particle size, zeta potential, long-term stability of SiO2-Gd2O3:Eu3+ should be investigated.
  2. The advantage of SiO2-Gd2O3:Eu3+ for PDT should be discussed in detail.
  3. The ROS production abilities of SiO2-Gd2O3:Eu3+ should be evaluated.
  4. How could SiO2-Gd2O3:Eu3+ be used for PDT?
  5. The evaluation of SiO2-Gd2O3:Eu3+ for PDT in living cells should be provided.
  6. In Figure 6 and Figure 8, SiO2-Gd2O3:Eu3+ seemed to be aggregated.
  7. In Figure 7, Gd and Eu could not be observed clearly.

Reviewer 2 Report

This manuscript describes how novel X-ray scintillating rare earth element nanoparticles can be prepared that are potentially useful for PDT research. All the relevant experiments have been carried out competently and the manuscript is generally well-written. The manuscript can be accepted after minor revisions. I urge the authors to consider the following points when preparing their revised manuscript:

(a) On line 76, the appropriate format of "D-glucuronic" should be considered carefully.

(b) The linewidth for the IR data in Figure 1 is too narrow and should be adjusted.

(c) Is the size of the subscripts in Figure 4 appropriate?

(d) In Figure 7 would a 50 nm rather than 100 nm scale size make the distributions easier to see? This issue is especially problematic for the bottom two panels. If the vertical colored bars provide information, this should be described in the caption.

(e) In the caption of Figure 8 and elsewhere the authors should sort out font size related issues.

(f) In Figure 11, a multiplication sign with spaces on either side should be used in 4.08 x 10-5.

Reviewer 3 Report

The manuscript is well organised and well presented, and physicochemical methods used in the experiments were adequate. The paper can be acceptable for publishing after a thorough revision:

  1. Introduction: during discussion of the using nanoparticles of various nature for cancer PDT, it is necessary to take into account the following topical articles: Davydenko et al. J. Mol. Liq., 2006, 127, 145-147; Scharff et al. Tumori, 2008, 94(2), 278-283; Prylutska et al. Exp. Oncol., 2010, 32, 29-32; Grebinyk et al. Free Radical Biology and Medicine, 2018, 124, 319-327.
  2. Results and Discussion: the authors showed that the created nanoparticles are not toxic against cancer cells (dosa-effect) and suppose to use them for PDT. When using any drugs in oncology, it is important to reduce their side (toxic) effect against normal cells. Therefore, the authors should additionally conduct a study of the cytotoxicity of these nanoparticles in relation to normal cells.